# POLICY IMPROVEMENT WITH STYLE-SPECIFIC DEMONSTRATIONS

## ABSTRACT

Proficient game agents with diverse play styles enrich the gaming experience and enhance the replay value of games. However, recent advancements in game AI based on reinforcement learning have predominantly focused on improving proficiency, whereas methods based on evolution algorithms generate agents with diverse play styles but exhibit subpar performance compared to RL methods. To address this gap, this paper proposes Mixed Proximal Policy Optimization (MPPO), a method designed to improve the proficiency of existing suboptimal agents while retaining their distinct styles. MPPO unifies loss objectives for both online and offline samples and introduces an implicit constraint to approximate demonstrator policies by adjusting the empirical distribution of samples. Empirical results across environments of varying scales demonstrate that MPPO achieves proficiency levels comparable to, or even superior to, pure online algorithms while preserving demonstrators' play styles. This work presents an effective approach for generating highly proficient and diverse game agents, ultimately contributing to more engaging gameplay experiences.

## 1 INTRODUCTION

Games benefit from having bots with varied proficiency and diverse play styles. Bots with different levels of proficiency accommodate a wider range of players, providing smoother gaming experiences (Climent et al., 2024; Romero-Mendez et al., 2023). In games featuring competition and cooperation, distinct agent styles provide value both as diversified opponents (Barros et al., 2023) and as adaptive partners for players with varied strategic preferences (Sweller, 1994; Chen, 2017). This is particularly evident in contemporary games that incorporate heterogeneous agents. From aggressive melee assassins to defensive support mages, characters in MOBA titles (Dota 2, League of Legends) and hero shooters (Valorant, Overwatch) possess distinct gameplay mechanics, and corresponding AI agents must replicate these stylistic nuances to provide immersive player experiences (Gao et al., 2023).

Advancement and application of Reinforcement Learning (RL) in game AI elevate agents' proficiency across various games. In traditional games, AlphaZero defeats top humans in chess, Shogi, and Go (Silver et al., 2017); PerfectDou (Yang et al., 2024) outperforms other AIs in DouDiZhu; AlphaHoldem (Zhao et al., 2022) beats human professionals in Texas Hold'em Poker. In computer games, AlphaStar (Vinyals et al., 2019) reaches grandmaster level in StarCraft II, OpenAI Five (Berner et al., 2019) defeats the Dota 2 world champion team, and JueWu (Ye et al., 2020) beats top esport players in The Honor of Kings. However, these methods prioritize reward maximization, and agents' play styles are not within their considerations.

The pursuit of diverse game-playing agents has attracted growing interest. While Quality-Diversity (QD) optimization (Lehman & Stanley, 2011) methods have been applied to generate varied behaviors (Canaan et al., 2019; Perez-Liebana et al., 2020), they often rely on predefined behavior descriptors and archive structures, which can limit their scalability and performance in complex domains like image-based games (Fuks et al., 2019; Chen et al., 2019; Badia et al., 2020). Meanwhile, within RL, population-based algorithms have emerged as a promising paradigm for fostering strategic diversity. These methods induce behavioral heterogeneity by optimizing agents towards divergent objectives, such as employing distinct risk preferences (Jiang et al., 2023), randomizing

reward functions (Tang et al., 2021), or directly maximizing a diversity metric (Parker-Holder et al., 2020) like the determinant of the population's behavioral embedding matrix.

While effective for discovering diverse behaviors, these population-based methods come at the significant computational cost of maintaining a large agent cohort and offer limited control over steering strategies toward a predefined style.

We therefore introduce a method that complements the population-based paradigm by efficiently "polishing" individual agents. Our approach tackles the distinct problem of *single-agent policy optimization with style preservation*: given suboptimal, stylized agents, we enhance their proficiency while preserving their play styles. To this end, we propose **M**ixed **P**roximal **P**olicy **O**ptimization (MPPO), a Learning from Demonstration (LfD) algorithm that leverages existing demonstrators to achieve this goal. Specifically, we employ two types of actors: on-policy actors improve policies and generalize learned behaviors to unseen states, while LfD actors imitate the demonstrators' policies. Samples generated by these actors are processed and trained using unified loss objectives. Through theoretical analysis, we prove that our method can monotonically improve the policy while satisfying implicit behavior cloning constraints. We test MPPO in three environments of varying complexity: Blackjack, Maze Navigation, and Mahjong. Empirical results demonstrate that MPPO is comparable or superior to baseline methods, achieving meaningful improvements in agent proficiency while retaining the unique characteristics of the original play styles. Notably, starting from suboptimal demonstrations, one of our Mahjong agents surpasses the top-ranked bot on Botzone's Elo ranking list (Zhou et al., 2018). The following are the key contributions of this paper:

- This paper proposes MPPO, a method that leverages data from suboptimal demonstrators to enhance policy proficiency while maintaining a relatively small distance from the demonstrators' policies.

- Theoretically, we demonstrate that our loss objectives are capable of monotonically improving the policy while guiding the student policy to imitate the demonstrator's.

- Enhancements to offline dataset collection and replay mechanisms enable the straightforward application of Monte Carlo-based advantage estimation, while reducing the storage footprint by at least 98% in our test scenarios.

- We present a suite of environments of varying scales as a benchmark to test agents' play style diversity and proficiency, with each environment accompanied by multiple stylized bots. In addition, we introduce a metric, $D_{policy}$, to quantify differences in play styles.

## 2 RELATED WORKS

Learning from Demonstration (LfD) is a broad research area that focuses on improving agents' learning efficiency by leveraging demonstration data to reduce the cost of agents' blind trial-and-error in environments. Based on whether methods integrate RL, LfD methods can be categorized into two categories: pure Imitation Learning (IL) (Zare et al., 2024), which learns policies solely through imitating experts' behaviors, and RL combined with demonstration learning, which utilizes demonstration data to accelerate the RL process.

Generative Adversarial Imitation Learning (GAIL) (Ho & Ermon, 2016) is a prominent IL method that considers IL problems as distribution matching, and it integrates adversarial training techniques (Goodfellow et al., 2014) to assign rewards for actions. Although this architecture solves challenges from sparse reward, students can hardly surpass demonstrator policies. POfD (Kang et al., 2018) and SAIL (Zhu et al., 2020) are GAN-based methods that build upon GAIL to address its limitations. However, these GAN-based methods suffer from inherent limitations of adversarial training, such as training instability and limited scalability in high dimensions (Brown et al., 2019).

Combining demonstration learning with reinforcement learning (RL) is appropriately termed Reinforcement Learning with Expert Demonstrations (RLED) (Piot et al., 2014). The core difference between RLED and IL is that the rewards are generated by the environment in RLED. The primary goal of methods (Chemali & Lazaric, 2015; Wagenmaker & Pacchiano, 2023; Hou et al., 2024) in this paradigm is to accelerate training or enhance performance using expert demonstrations, which aligns with our research focus. Additionally, our approach aims to **preserve the demonstrators' behavioral styles.**

Under the paradigm of RLED, existing approaches diverge in how they integrate offline data. Some methods employ a sequential strategy, pre-training on demonstrator data before fine-tuning with online interactions (Nair et al., 2021; Nakamoto et al., 2024), while others leverage offline and online data concurrently throughout training (Hester et al., 2017; Ball et al., 2023).

Since our primary objective is to achieve policy improvement while preserving the play style of the original demonstrators, we adopt the latter approach, which allows continuous guidance from demonstration data during policy optimization. Within this concurrent learning setting, a number of methods have been tailored to specific action space structures. For instance, DDPGfD (Vecerík et al., 2017) and RLPD (Ball et al., 2023) are designed for continuous control domains, whereas DQfD (Hester et al., 2017) represents a leading approach tailored to discrete action spaces, a common setting in game environments that motivates our approaches.

DQfD integrates deep Q-learning (Mnih et al., 2013) with demonstration data by combining multi-step temporal difference (TD) and supervised losses, aiming to address state distribution bias and accelerate the convergence of the learning process. While our method aligns with DQfD in terms of application settings, there are key distinctions beyond the differences in their backbone algorithms. Specifically, DQfD employs explicit supervised losses to guide student policies; instead, we utilize implicit soft constraints to guide student policies through demonstration data filtering—this is rooted in our prior awareness that demonstrations may be suboptimal. Furthermore, we incorporate multi-step TD into Generalized Advantage Estimation (GAE) (Schulman et al., 2015), which significantly simplifies the algorithm's architecture.

## 3 PRELIMINARIES

We formulate game environments as standard Markov Decision Processes (MDP) $\mathcal{M} = \langle \mathcal{S}, \mathcal{A}, R, \mathcal{P}, \gamma, \rho_0 \rangle$, where $\mathcal{S}$ and $\mathcal{A}$ are the observable state space and the action space, respectively, $R(s, a)$ represents the reward function, and $\gamma \in (0, 1)$ is the discount factor. $\rho_0$ is the initial state distribution. Policy $\pi(a_t|s_t)$ is defined as the distribution of actions conditioned on states at step $t$, where $s_t \in \mathcal{S}$ and $a_t \in \mathcal{A}$. $\mathcal{P}(s'|s, a)$ is the transition distribution of taking action $a$ at observable state $s$. Both randomness from environment dynamics and randomness due to unobservable state information are attributed to $\mathcal{P}$ in the formulation. The trajectory $\tau = \{s_t, a_t\}_{t=0}^T$. The performance measure of policy is defined as $J(\pi) = \mathbb{E}_\pi \sum_{t=0}^T \gamma^t R(s_t, a_t))$. Then, the value function, state-value function, and the advantage function can be defined as $V_\pi(s) = J(\pi|s_0 = s)$, $Q_\pi(s, a) = J(\pi|s_0 = s, a_0 = a)$, and $A_\pi(s, a) = Q_\pi(s, a) - V_\pi(s)$, respectively.

**Theorem 1.** (Kakade & Langford, 2002) Let the discounted unnormalized visitation frequencies as $\rho_\pi(s) = \sum_{t=0}^T \gamma^t P(s_t = s|\pi)$, and $P(s_t = s|\pi)$ represents the probability of the t-th state equals to $s$ in trajectories generated by policy $\pi$. For any two policies $\pi$ and $\pi'$, the performance difference $J_\Delta(\pi', \pi) \triangleq J(\pi') - J(\pi)$ can be measured by:

$$J_\Delta(\pi', \pi) = \mathbb{E}_{s \sim \rho_{\pi'}(\cdot), a \sim \pi'(\cdot|s)}[A_\pi(s, a)]. \quad (1)$$

This theorem implies that improving policy from $\pi$ to $\pi'$ can be achieved by maximizing equation 1. From this theorem, Trust Region Policy Optimization (TRPO) (Schulman et al., 2017a) and Behavior Proximal Policy Optimization (BPPO) (Zhuang et al., 2023) are derived, which can guarantee the monotonic performance improvement for online and offline settings, respectively.

**Metric for Play Style Distance**   To quantify the similarity between play styles, metrics are defined based on action distributions. Lin et al. (2024) used the 2-Wasserstein distance ($W_2$) (Vaserstein, 1969) to measure the distance between play styles; however, $W_2$ is computationally expensive as a metric. More critically, its value depends on an arbitrary embedding of actions into a metric space to define pairwise distances. This makes results inconsistent and difficult to interpret, as different embeddings yield different distances for the same policies. Thus, we use total variational divergence, denoted as $D_{policy}$, to overcome $W_2$'s drawbacks and measure play style distances. Defined in equation 2, it quantifies action-level play style discrepancies.

$$D_{policy} = \mathbb{E}_{s \in S} \frac{1}{2} \sum_{a \in A} |\pi_1(a|s) - \pi_2(a|s)| \quad (2)$$

## 4    PROPOSED ALGORITHM

Our algorithm integrates data from both online environmental interactions and offline demonstration datasets. We therefore term it **M**ixed **P**roximal **P**olicy **O**ptimization (MPPO).

In this section, we establish two fundamental theoretical properties of MPPO: 1) the policy improves monotonically, and 2) the policy remains proximal to the demonstration policy throughout the learning process. Subsequently, we provide the pseudocode and implementation details of MPPO.

We denote collections of demonstration trajectories as $D = \{\tau_1, \tau_2, ..., \tau_N\}$ and the policy at some point during the learning process as $\pi_k$. Our approach to policy improvement builds upon Theorem 1. However, directly optimizing equation 1 is intractable due to its dependence on the unknown state distribution $\rho_{\pi'_k}(s)$ of the new policy. Standard online RL methods, such as TRPO, address this by approximating $\rho_{\pi'_k}(s)$ with $\rho_{\pi_k}(s)$, the state distribution of the current policy $\pi_k$, thereby guaranteeing monotonic improvement under on-policy data.

In our setting, which incorporates off-policy demonstration data, we propose a mixed state distribution. We sample a fraction $\beta$ of state-action tuples from the demonstration dataset $D$, and the remainder $1 - \beta$ from the current policy $\pi_k$. This leads to the empirical state distribution $\rho_{mix}(s) = \beta\rho_D(s) + (1-\beta)\rho_{\pi_k}(s)$, where $\rho_D(s)$ is the state visitation distribution in the demonstration data. Substituting this mixed distribution into equation 1 yields the following surrogate objective:

$$\hat{J}_\Delta(\pi, \pi_k) = \mathbb{E}_{s \sim \beta\rho_D(\cdot) + (1-\beta)\rho_{\pi_k}(\cdot), a \sim \pi(\cdot|s)}[A_{\pi_k}(s,a)]$$

$$= \beta\mathbb{E}_{s \sim \rho_D(\cdot), a \sim \pi(\cdot|s)}A_{\pi_k}(s,a) + (1-\beta)\mathbb{E}_{s \sim \rho_{\pi_k}(\cdot), a \sim \pi(\cdot|s)}A_{\pi_k}(s,a) \tag{3}$$

The monotonic improvement guarantee for the second term in equation 3, which handles on-policy data, is well-established by TRPO. For the first term, which utilizes off-policy demonstration data, recent work on BPPO provides analogous theoretical guarantees, ensuring improvement under static dataset constraints. We leverage these prior theoretical results. Since our objective is a linear combination of the objectives from TRPO and BPPO, the monotonic improvement property is preserved for the combined objective $\hat{J}_\Delta$.

For practical implementation and stability, we adopt the clipped surrogate objective from Proximal Policy Optimization (PPO) (Schulman et al., 2017b), which provides a first-order approximation to the constrained optimization problems solved by TRPO and BPPO. This yields our final practical loss function:

$$L^{MPPO} = \beta\mathbb{E}_{s \sim \rho_D(\cdot)}[\min(rA_{\pi_k}(s,a), \text{clip}(r, 1-\epsilon, 1+\epsilon)A_{\pi_k}(s,a))]$$

$$+ (1-\beta)\mathbb{E}_{s \sim \rho_{\pi_k}(\cdot)}[\min(rA_{\pi_k}(s,a), \text{clip}(r, 1-\epsilon, 1+\epsilon)A_{\pi_k}(s,a))] \tag{4}$$

where $\epsilon$ restricts new $\pi'_k$ from deviating from $\pi_k$, and $r = \frac{\pi'_k(a|s)}{\pi_k(a|s)}$ serves a dual purpose:

- **For the on-policy samples**, it acts as the **probability ratio**, quantifying the change in the policy probability for a given action and forming the basis of the PPO clipping objective.

- **For the off-policy demonstration samples**, it functions both as the **probability ratio** within the PPO objective and as the **importance sampling** mechanism to correct for the distribution shift between the behavior policy and the current policy. This formulation provides an implicit constraint that anchors the updated policy $\pi'_k$ to its immediate predecessor $\pi_k$, which, due to small PPO updates, remains in the neighborhood of the teacher's state-action distribution. While BPPO introduces a decaying clipping ratio to further mitigate distribution shift, we employ a small, fixed clipping value as a more simple and robust alternative.

The mathematical unity of this term enables the seamless integration of offline and online data into a consistent objective function. Thus, MPPO is guaranteed to improve the policy monotonically. Next, we need to ensure the similarity between the student policy and the demonstration policy.

**Theorem 2.** $\pi_T$ is the teacher's policy, and $\pi_S$ is the student's policy. Given $D = \{\tau_1, \tau_2, ... \tau_N\}$, where each $\tau_i$ is sampled with $\pi_T$, the play style distance $D_{policy}(\pi_S, \pi_T)$ is non-increasing and is systematically guided toward the demonstrator's policy under the influence of the offline component of MPPO, provided that $\forall (s_t, a_t) \in \tau_i$, the value $A_{\pi_T}(s_t, a_t)$ is positive[1].

Theorem 2 ensures that the policy is regularized to limit its divergence from the demonstrator's policy. The MPPO objective (Eq. equation 4) primarily drives performance improvement, which may naturally lead the policy away from the teacher. The implicit constraint in Theorem 2 acts as a regularizer, anchoring the policy to the demonstrator's behavioral style. The hyperparameter $\beta$ governs the equilibrium between these dual objectives of proficiency and style consistency.

We have also modified the storage and replay mechanisms for the demonstration data in MPPO. Existing LfD and offline algorithms typically provide datasets as collections of $(s, a, r)$ or $(s, a, r, s', a')$ tuples. These methods either rely on 1-step TD estimation or require full processing and computation of entire trajectories to recover episodic information, which is necessary for multi-step TD or Monte Carlo-based advantage estimation. Additionally, a drawback of existing datasets is that complete state information must be stored, which often results in large storage footprints.

We collect demonstration trajectories $\tau_i$ by recording the environment initialization seed and action sequences $\{a_t\}_{t=0}^{T}$. During the training phase, complete episodes can be recovered by replaying these action sequences without modifying other online RL components. This enables full-episode advantage estimation methods, such as GAE, to be applied to both online and offline samples.

Building on the aforementioned results, we derive a practical algorithm. During training, two types of sampling actors are instantiated: 1) on-policy actors, which collect training data through environment interactions using the latest student policy as described in the original PPO algorithm; and 2) LfD actors, which reproduce demonstrator data from the demonstration trajectories. To satisfy the conditions of Theorem 2, we filter trajectories from the dataset $D$, retaining only those with positive total returns for policy optimization. In particular, LfD actors initialize the game environment using the recorded seed and feed the action sequences to generate complete $(s, a, r, v, adv)$ tuples as training samples. Here, $v$ denotes the state value estimated by the critic network, and $adv$ represents the GAE advantage calculated from $r$ and $v$. This approach ensures that data collected by both types of actors can be processed uniformly before being fed into the replay buffer for policy improvement.

To summarize, on the actor side, we modified the collection and replay mechanisms for offline data, enabling accurate full-episode advantage estimates to be readily applied to offline samples. On the learner side, theoretical results have established that MPPO can monotonically improve policies in both online and offline reinforcement learning settings. Additionally, MPPO's behavior cloning constraint is implicitly defined between the demonstration policy $\pi_T$ and the student policy $\pi_S$ via data filtering. The pseudocode for MPPO is shown in Algorithm 1.

## 5 EXPERIMENTS

In this section, we aim to investigate two key questions: 1) **whether MPPO can meaningfully enhance agents' game proficiency beyond suboptimal demonstrations**; and 2) **whether the improved agents can retain their game styles.**

We adopt the IMPALA architecture (Espeholt et al., 2018) for our experiments. At the end of each update step, the learner sends the latest model parameters to all actors, which then update their parameters before initiating new episodes. For MPPO agents in each environment, we adjusted the ratio of on-policy actors and LfD actors such that **demonstration data accounts for approximately 5%** ($\beta = 0.05$) of the total incoming data.

We conduct experiments across three environments of varying scales: Blackjack, Maze, and MCR Mahjong. For each environment, we provide multiple suboptimal demonstrators, from which approximately 30K demonstration trajectories with positive outcomes are collected per demonstrator. To compare storage footprints, we collect datasets in both our proposed format and the traditional format. Results show that our format reduces storage usage by 98%[2].

---

[1]The conclusion holds solely based on the positivity of $A_{\pi_T}(s_t, a_t)$, regardless of how $A_{\pi_T}$ is defined or interpreted. The proof of the theorem is included in Appendix A

[2]Blackjack: 1.13MB VS 84MB, Maze: 120MB VS 18.3GB, Mahjong: 155MB VS 40GB.

---

**Algorithm 1** Mixed Proximal Policy Optimization

---

**Input**: Collections of Demonstrations: $D = \{\tau_1, \tau_2, ...\tau_N\}$,
Actor policy: $\pi_\theta$, Critic policy: $V_\phi$, Demo Indicator: $d$

1: **for** n=1,2,... **do**
2:    **if** Demo Indicator $d \sim U(0,1) < \beta$ **then**
3:       sample $\tau_i = \{(s_t, a_t)\}_{t=0}^T \sim D$
4:       Initialize environment
5:       **for** t=0,1,...,T **do**
6:          retrieve action $a_t \in \{a_t\}_{t=0}^T$ from $\tau_i$
7:          estimate state value with $V_\phi$
8:          send trajectories with positive returns to learner
9:       **end for**
10:   **else**
11:       Randomly initialize environment
12:       **for** t=0,1,...,T **do**
13:          sample action $a_t \sim \pi_\theta$
14:          estimate state value with $V_\phi$
15:          send all trajectories to learner
16:       **end for**
17:   **end if**
18:   calculate advantage with GAE
19:   update $V_\phi$ and $\pi_\theta$ with MPPO loss equation 4
20: **end for**

---

Student agents are evaluated in terms of their game proficiency and their play style distance to their corresponding demonstrators. All experiments are repeated 5 times with different random seeds, and detailed experimental configurations for each environment are provided in Appendix C. Our anonymized data and codes are available at https://github.com/AMysteriousBeing/MPPO.

### 5.1 BLACKJACK

**Settings** Blackjack is a single agent stochastic game, and its rules and settings closely follow the descriptions in Sutton & Barto (2018). For players, the goal is to attain a hand closer to 21 than the dealer's without exceeding this value. We provide a rule-based Blackjack Bot A, whose policy is represented by the red dashed lines in Figure 1. We trained MPPO student agents using demonstration data from Bot A and compared their win rates with those of PPO agents, the optimal policy, and Bot A's policy. A total of 15,000 fixed seeds are used to test the win rates of the rule-based bots and student agents.

**Results** As shown in Table 1A, the win rates of MPPO agents are comparable to those of the optimal policy and PPO agents, yet significantly higher than those of Bot A. Given the tractable state space of Blackjack, we can analyze agents' game proficiency by examining their policy decision boundaries. As illustrated in Figure 1, MPPO agents consistently converge between the optimal policy and the demonstrator's policy. This indicates that, in Blackjack, MPPO agents can surpass the game proficiency of the demonstrator while maintaining its play styles, an observation further supported by Table 1B. Specifically, MPPO agents' policies are more distant from the optimal policy and closer to Bot A's policy compared to PPO agents' policies.

### 5.2 MAZE NAVIGATION

**Settings** The maze environment is a deterministic pathfinding task set in a 19×19 random grid world, where a valid path from the entrance to the exit is guaranteed. At each step within the maze, agents move in a single direction until they encounter a fork or a wall. The environment terminates under two conditions: if the agent reaches the exit, it receives a positive reward; if the step limit of 80 is exceeded, no reward is given. Within the maze, the agent can observe 5 adjacent grids around its current position, and an example of a random maze is provided in Figure 2.

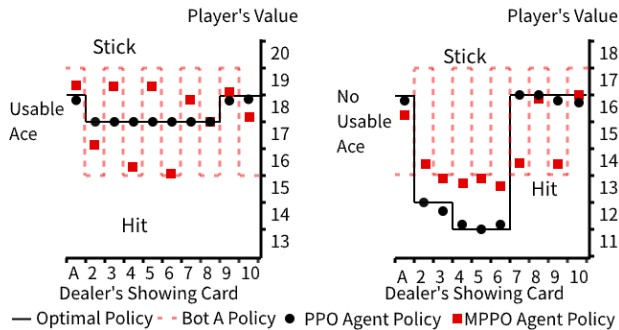

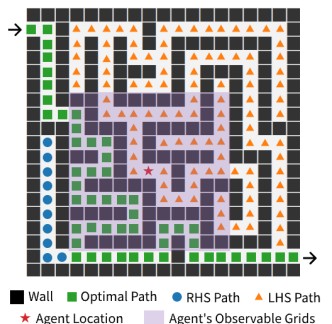

Figure 1: Visualization of Decision Boundaries Learned by PPO and MPPO Agents.

Figure 2: Example of a random maze.

Table 1: Blackjack A). Win rates and B). $D_{policy}$ results

| A | Optimal | Bot A | PPO Agent | MPPO Agent |
|---|---|---|---|---|
| | 43.26 | 40.84 | 43.40±0.17 | 42.82±0.08 |

| B | Optimal Policy | Bot A |
|---|---|---|
| PPO | 0.042±0.008 | 0.259±0.006 |
| MPPO | 0.135±0.004 | 0.150±0.011 |

Table 3: Mahjong's Bot Information

| Bot Name | UUID | Ranking | Elo Score |
|---|---|---|---|
| Baseline | 5eb7...123e | 1 | 1328.76 |
| Bot A | 5fdf...5837 | 17 | 1240.50 |
| Bot B | 627e...c460 | 161 | 1128.66 |
| Bot C | 5ecc...eb73 | 266 | 980.51 |

Table 2: Maze A). Success Rates and B). $D_{policy}$ Results.

| A)Agents | Win Rate% | Avg Step |
|---|---|---|
| Optimal | 100.00 | 25.224 |
| Bot A | 92.40 | 53.450 |
| Bot B | 89.00 | 55.714 |
| PPO | 99.64±0.12 | 27.230±0.147 |
| MPPO A | 99.52±0.31 | 27.649 ±0.440 |
| MPPO B | 99.04±0.81 | 28.104 ± 0.649 |

| B)Agents | Optimal | Bot A | Bot B |
|---|---|---|---|
| PPO | .057±.002 | .509±.002 | .492±.001 |
| MPPO A | .084±.001 | .471±.013 | .530±.008 |
| MPPO B | .076±.004 | .521±.010 | .481±.009 |

For maze-navigating bots, we provide Maze Bot A, a right-hand search (RHS) bot, and Maze Bot B, a left-hand search (LHS) bot. The RHS and LHS bots implement right-hand and left-hand wall-following behaviors, respectively. For the maze environment, we use success rates and average steps to quantify agents' proficiency. A run is considered successful if the agent reaches the exit within the step limit. We evaluated agents in 500 unseen mazes.

**Results** As shown in Table 2A, the success rates of MPPO agents are comparable to those of PPO agents, yet MPPO agents require slightly more steps on average to exit the mazes. In contrast to their demonstrators, MPPO agents exhibit significantly higher success rates, with their average steps reduced by approximately 20.

For the $D_{policy}$ metric, Table 2B demonstrates that MPPO achieves meaningful policy improvement while preserving the navigation styles of its demonstrators. This can be explained by the strategies learned by the agents. The MPPO A agent, while converging to a near-optimal policy, retains a stylistic preference for right-hand turns inherited from its demonstrator (Maze Bot A). Analytically, this is reflected in its policy logits, which show a stronger propensity for right turns compared to the PPO agent when facing ambiguous states (as the environment is partially observable). Behaviorally, this bias manifests as more occasional detours to the right in certain maze configurations, which directly accounts for the slightly increased average path length in Table 2A. The same principle explains the results for MPPO B, which exhibits a symmetric preference for left-hand turns.

### 5.3 MAHJONG

**Settings** Mahjong is a multi-player game with imperfect information. The complexity of imperfect-information games can be quantified by information sets (info sets), which refer to game

states that players are unable to differentiate based on their observations. Mahjong features around $10^{121}$ info sets, with the average size of each set estimated at $10^{48}$, a complexity vastly exceeding that of Heads-Up Texas Hold'em, where the average info set size is roughly $10^3$ (Lu et al., 2023).

The game is played with a set of 144 tiles. Each player begins with 13 tiles, which are only observable by themselves. They take turns to draw and discard a tile until one completes a winning hand with a 14th tile. Our environment adopts the Mahjong Competition Rules (MCR) variant, which contains 81 different scoring patterns. The details of the MCR are provided in Appendix B.

For the Mahjong environment, we additionally analyze the distribution of winning patterns between agents, as these patterns reflect the strategies employed by the winners during the game. We denote the distance between winning pattern distributions as $D_{target}$, defined in equation 5. Here, $p$ denotes an MCR pattern, $P$ represents the set of all patterns, and $\pi_i(p)$ refers to the probability that an agent following policy $\pi_i$ wins with pattern $p$. Compared to $D_{policy}$, $D_{target}$ provides a more straightforward measure of play style, enabling us to examine whether micro-level play styles indeed influence macro-level strategies.

$$D_{target} = \frac{1}{2} \sum_{p \in P} |\pi_1(p) - \pi_2(p)| \qquad (5)$$

The MCR Mahjong bots are selected from Botzone (Zhou et al., 2018), an online platform for AI in games. As shown in Table 3, the demonstrators, specifically Bot A, B, and C, are deliberately chosen from different performance ranges. For reference, currently there are over 600 bots on the platform, and Elo scores range from 460 to 1328. To accelerate training, MPPO and PPO agents (A, B, and C) are initialized using behavior cloning checkpoints derived from Bot A, B, and C, respectively[3].

All student agents are evaluated for game proficiency against the Baseline bot every 12 hours. The win rates of the demonstrator bots are calculated directly from Botzone's historical Elo data. Meanwhile, the win rate of each student agent is determined by testing the agent against the baseline bot over 512 games. We use the final checkpoints of the agents to calculate $D_{policy}$. For the action distributions of the demonstrator bots, we collect $(s, a)$ pairs from 100 trajectories **not** used in training, and we set $p(a|s) = \mathbf{1}_{a=a_i}, \forall a_i \in A$. We exclude states $s$ with only one legal action and feed the remaining states into the agents' models. Similarly, for $D_{target}$, the winning pattern distribution $\pi(p)$ of the demonstrator bots is calculated directly from their historical game data, whereas the $\pi(p)$ of student agents is derived from policy evaluation runs using 20,000 fixed seeds.

Table 4: Mahjong A). Win Rates and B). $D_{policy}$ Results.

| A)Win Rate VS Base | Teacher Bot | MPPO Agents | PPO Agents |
|---|---|---|---|
| Bot A | 43.67 | **51.05**±**1.43** | 36.72±3.11 |
| Bot B | 39.82 | 46.17±1.72 | 34.96±2.57 |
| Bot C | 37.05 | 42.42±3.66 | 33.32±1.41 |
| B)$D_{policy}$ | | MPPO | PPO |
| Bot A | | 0.297±.016 | 0.678±.027 |
| Bot B | | 0.318±.007 | 0.691±.013 |
| Bot C | | 0.279±.020 | 0.772±.027 |

Table 5: $D_{target}$ between teachers and students. Values between student and demonstrator pairs are highlighted.

| $D_{target}$ | Bot A | Bot B | Bot C |
|---|---|---|---|
| Bot A | 0 | .023 | .071 |
| PPO A | **.195**±**.003** | .208±.003 | .215±.004 |
| MPPO A | **.037**±**.008** | .047±.009 | .077±.002 |
| Bot B | .023 | 0 | .063 |
| PPO B | 0.201±.007 | **.214**±**.007** | .221±.008 |
| MPPO B | 0.047±.007 | **.039**±**.005** | .068±.004 |
| Bot C | .071 | 0.063 | 0 |
| PPO C | .191±.006 | .204±.007 | **.212**±**.007** |
| MPPO C | .086±.002 | .076±.010 | **.047**±**.012** |

**Results** The performance of agents against the baseline is presented in Figure 3. MPPO agents quickly surpass their demonstrators, and Bot A student agents defeat the baseline at the end of training. We recorded the best performance of each student agent across all runs, and the results are summarized in Table 4A. For reference, the champion bot from the IJCAI 2024 Mahjong AI Competition ranks 33rd in Botzone's Elo rankings. This indicates that MPPO agents can outperform

---

[3]This warm-start is employed solely to reduce wall-clock time and resource consumption and is not a methodological prerequisite, as shown in Appendix G.

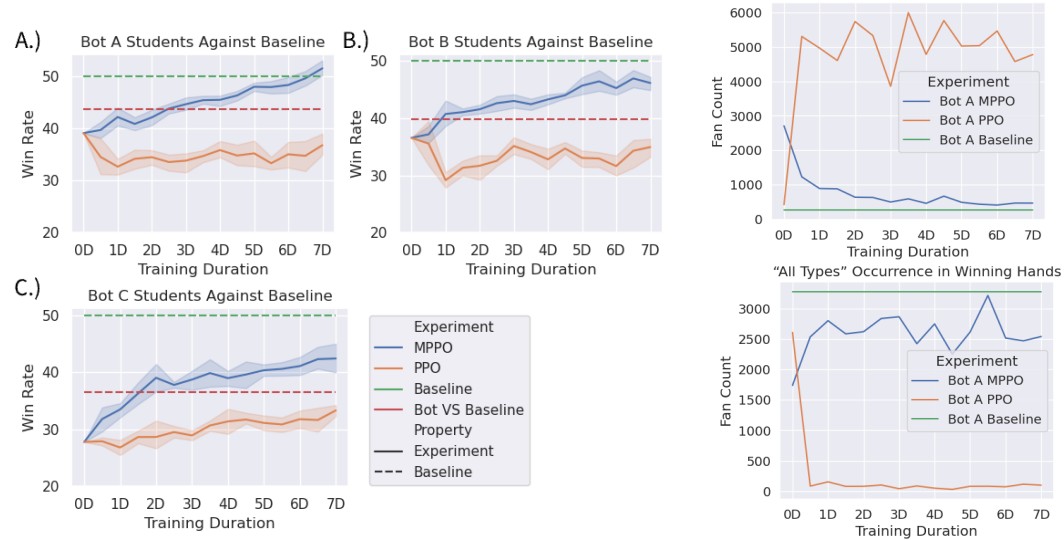

Figure 3: Bots' win rates against the Baseline. Red dashed lines are the demonstrators' win rates against Baseline. Shaded areas are 95% confidence intervals.

Figure 4: Occurrence of selected patterns.

top-tier bots using suboptimal demonstrations. As shown in Table 4B, the $D_{policy}$ values of student agents relative to their demonstrator bots indicate that MPPO agents have action distributions significantly closer to their demonstrator bots than PPO agents.

In Table 5, the $D_{target}$ values between MPPO student agents and their corresponding demonstrators are consistently the lowest, confirming the observation from $D_{pocliy}$. Additionally, Bot A and Bot B have relatively close target distributions, and this proximity in target preference is also inherited by MPPO agents A and B.

In the calculation of $D_{target}$, we observe that PPO agents rapidly lose the ability to achieve some patterns while focusing on several others. In contrast, MPPO agents retain the ability to achieve most patterns, a capability preserved by the demonstration trajectories, as illustrated in Figure 4. This phenomenon explains why the $D_{target}$ values of PPO agents are significantly higher than those of MPPO agents.

Now we can address the questions posed at the beginning of this section. MPPO exhibits a strong ability to surpass the proficiency of demonstrators and, in some cases, even outperform PPO. By comparing $D_{policy}$ values among PPO agents, MPPO agents, and bots, we conclude that MPPO agents imitate the play styles of their demonstrators at the action level. Further analysis of $D_{target}$ in Mahjong confirms that such action-level style similarities extend to the strategy level, for example, target selection in MCR Mahjong, and that MPPO agents thus retain their demonstrators' play styles.

# 6 ABLATION AND COMPARATIVE STUDY

To analyze the impact of different components of the MPPO algorithm, we conducted ablation studies using Bot A's trajectories across each environment. Table 6 summarizes the win rates and $D_{policy}$ values between the agent groups and their corresponding demonstrators[4].

For the **2x Demo** and **0.5x Demo** experiments, we doubled and halved the value of $\beta$, respectively, to analyze the impact of the demonstration data ratio. As expected, a higher ratio of demonstration data leads to a lower $D_{policy}$. The proportion of demonstration data also affects the final proficiency of the agents. In each environment, the proficiency metrics of MPPO agents peak at different ratios, indicating that different environments correspond to unique optimal ratios of demonstration data.

---

[4]The learning curves are presented in Appendix C.

Table 6: Summarized ablation and comparison study results for A) Win rates and B) $D_{policy}$ with each environment's Bot A. Ablation and comparative study results are separated by a horizontal line. Highest Win rates and lowest $D_{policy}$ values in ablation study results are highlighted.

| Method | A) Blackjack | Maze | Mahjong | B) Blackjack | Maze | Mahjong |
|---|---|---|---|---|---|---|
| PPO | 43.40±0.17 | **99.64±0.12** | 36.72±3.11 | .259±.006 | .509±.002 | .678±.027 |
| MPPO Ref | 42.82±0.08 | 99.52±0.31 | **51.05±1.43** | .150±.011 | .471±.013 | .297±.016 |
| 2x Demo | 42.08±0.06 | 99.60±0.20 | 48.09±1.86 | **.093±.007** | **.457±.018** | **.287±.006** |
| 0.5x Demo | 42.93±0.12 | 99.12±0.99 | 46.84±1.97 | .186±.003 | .492±.012 | .324±.007 |
| All Data | **43.62±0.04** | 94.24±0.81 | 42.30±3.65 | .294±.003 | .572±.016 | .775±.057 |
| TD(0) Adv | 43.31±0.28 | 99.40±0.69 | 17.34±0.74 | .218±.004 | .523±.003 | .727±.042 |
| GAIL | 25.63±1.21 | 92.40±0.00 | 3.32±0.58 | .501±.010 | 2e−6 ± 0 | .784±.044 |
| SAIL | 38.50±0.01 | 94.52±0.70 | 19.73±3.30 | .495±.006 | .540±.017 | .690±.025 |
| DQfD | 42.26±0.29 | 87.20±2.62 | 15.43±1.38 | .411±.003 | .659±.001 | .793±.004 |
| PPOfD | 42.89±0.15 | 99.40±0.20 | 41.52±2.59 | .231±.011 | .347±.009 | .335±.019 |

For the **All Data** experiments, we regenerate all datasets to include all trajectories for demonstration. This violates the prerequisite condition $A_\pi(s_t, a_t) > 0$ in Theorem 2, reducing the entire algorithm to online PPO where a fraction of the actors sample from fixed seed environments with a fixed policy. In this setting, $D_{policy}$ values are high in all environments, and the win rates vary by environment.

For the **TD(0) Adv** experiments, we replace GAE with 1-step TD advantage, an approach commonly adopted in existing offline RL and LfD methods. This weakens the prerequisite condition $A_\pi(s_t, a_t) > 0$ in Theorem 2, since 1-step TD responds slowly to the final reward. Consequently, we observe higher $D_{policy}$ values in all settings. While TD(0) performs well in Blackjack and Maze, it struggles in Mahjong, a more complex environment with long-horizon decision sequences, as it fails to leverage all future information.

For the comparative study, we compare MPPO with other LfD and IL methods: GAIL, SAIL, and DQfD. As shown in Table 6A, GAIL and SAIL perform well in Maze, a 2D state-space environment, yet struggle in Mahjong, where defining similarity between state-action pairs is challenging. This aligns with the findings of Brown et al. (2019), which note that adversarial-based IL methods do not scale effectively to high-dimensional scenarios.

DQfD also performs poorly in Mahjong: its training trajectories for the game exhibit the same low-entropy characteristics as those of the MPPO algorithm. To eliminate the influence of backbone algorithms and differences in action-sampling strategies, we ported DQfD's explicit supervised loss to MPPO, creating *PPOfD*. In essence, PPOfD differs from MPPO solely in the mechanism by which it encourages students to imitate demonstrators. PPOfD outperforms DQfD across all environments; it is comparable to MPPO in Blackjack and Maze but lags significantly behind MPPO in Mahjong. This indicates that our implicit behavior cloning constraint is more adaptable to diverse environments than explicit loss functions.

Regarding play styles, as shown in Table 6B, MPPO is the only method that meaningfully maintains low $D_{policy}$ values while improving agent proficiency across all environments.

## 7 CONCLUSION

In this paper, we tackle the dual objectives of proficiency and diversity in game-playing agents through MPPO, a method that enhances the proficiency of suboptimal agents while preserving their play styles. Through theoretical analysis, MPPO unifies the loss objectives for both online and offline samples, and implicitly guides student agents toward the demonstrators' policies by adjusting the empirical distribution of samples. Our experiments show that MPPO matches or even outperforms the pure online baseline (PPO) in proficiency, while preserving demonstrators' game styles by closely aligning with their policy distributions. Looking ahead, we aim to extend our method to continuous action domains. In addition, exploring trajectory-level metrics, such as state visitation distributions, presents a promising path for a richer characterization of behavioral style. We expect this work to contribute to more engaging gameplay and a more diverse agent ecosystem.

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

## A    PROOF OF THEOREM 2

*Proof.* We study the effect of MPPO's offline objective when clipping is not activated. Define $\mathbf{1}_{a=x}$ as 1 if action $a = x$, else 0, then $\pi_T(a|s_t) = \mathbf{1}_{a=a_t}$ Then $\forall (s_t, a_t) \in \tau_i$, the next iteration policy $\pi'_S(a_t|s_t) = \pi_S(a_t|s_t) + \alpha \frac{\nabla \pi_S(a_t|s_t)}{\pi_T(a_t|s_t)} A_{\pi_T}(s_t, a_t)$, where $\alpha$ is a constant. The change in $2D_{policy}(\pi_S, \pi_T) = \sum_a |\pi_T(a|s_t), \pi_S(a|s_t)|$ is:

$$\sum_a (|\pi_T(a|s_t), \pi'_S(a|s_t)| - |\pi_T(a|s_t), \pi_S(a|s_t)|) = \sum_{a=a_t} (\pi_S(a|s_t) - \pi'_S(a|s_t)) + \sum_{a \neq a_t} (\pi'_S(a|s_t) - \pi_S(a|s_t))$$

$$= \pi_S(a_t|s_t) - \pi'_S(a_t|s_t) + \pi_S(a_t|s_t) - \pi'_S(a_t|s_t) = -2\alpha \frac{\nabla \pi_S(a_t|s_t)}{\mathbf{1}_{a=a_t}} A_{\pi_T}(s_t, a_t)$$

$-\alpha \frac{\nabla \pi_S(a_t|s_t)}{\mathbf{1}_{a=a_t}} A_{\pi_T} < 0$ as $\alpha > 0$, thus $D_{policy}(\pi_S, \pi_T)$ decreases as training progresses if $A_{\pi_T}$ is positive. $\square$

## B    MCR MAHJONG ENVIRONMENT DESCRIPTION

Mahjong is a four-player tile-based tabletop game involving imperfect information. The complexity of imperfect-information games can be quantified by information sets, which refer to game states that players are unable to differentiate based on their own observations. The average size of information sets in Mahjong is approximately $10^{48}$, rendering it a considerably more complex game to solve compared to Heads-Up Texas Hold'em, where the average size of information sets is around $10^3$. To enhance the readability of this paper, we highlight the terminologies used in Mahjong with **bold texts**, and we differentiate scoring patterns with *italicized texts*.

In Mahjong, there are 144 tiles, as depicted in Figure 5A. Despite the existence of numerous rule variants, the general rules of Mahjong remain the same. At a broad level, Mahjong is a pattern-matching game. Each player starts with 13 tiles that are only visible to themselves, and they take

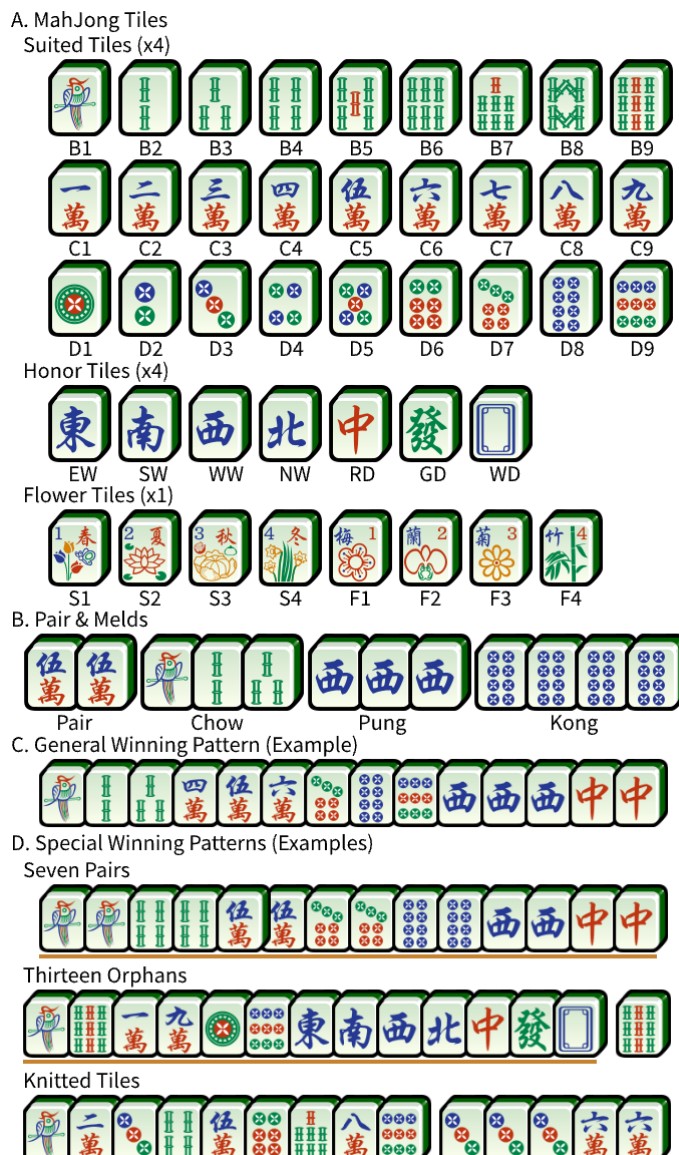

Figure 5: Basics of Mahjong. A). All the Mahjong tiles. There are four identical copies for each tile. B). Examples of Chow, Pung, and Kong. Only suited tiles are available for Chow. C). Example of the GWP. D). Examples of special winning patterns, the special patterns are seperated and underscored.

turns to draw and discard one tile until one completes a winning pattern with a 14th tile. The general winning pattern (GWP) of 14 tiles consists of four **melds** and a **pair**, as shown in Fig. 5C. A **meld** can be in the form of **Chow**, **Pung**, or **Kong**, as shown in Fig. 5B. Besides drawing all the tiles by themselves, players have the option to take the tile just discarded by another player instead of drawing one to form a **meld** or declare a win.

### B.1 OFFICIAL INTERNATIONAL MAHJONG

Official International Mahjong, also known as Mahjong Competition Rules (MCR), is a Mahjong variant aiming to enhance the game's complexity and competitiveness while weakening its gambling nature. It specifies 81 scoring patterns, which range from 1 to 88 points. In addition to forming the general winning pattern (GWP), players must accumulate at least 8 points by matching at least one scoring pattern in order to declare a win. Among the 81 patterns, 56 are highly valued and are

referred to as major patterns, since most winning hands usually include at least one of them. Some special patterns do not adhere to the GWP, such as *Seven Pairs*, *Thirteen Orphans*, and *Knitted Straight*, as illustrated in Fig 5D.

The final scores of each player depend on the winner's fan value and the provider of the 14th winning tile. Specifically, if the winner makes a winning hand of $x$ fans by drawing a tile themselves, they receive $8 + x$ points from the other three players. Instead, if the 14th winning tile comes from another player, either discarded or added to the promoted pung, the winning player receives $8 + x$ points from the provider of this tile, and only 8 points from the other two players.

### B.2 MCR AS AN ENVIRONMENT

As an environment, MCR exhibits several unique characteristics that pose challenges to algorithms.

First, the 8-point-to-win rule of MCR adds an additional requirement to the hand patterns. This requirement excludes many hand patterns that would otherwise be valid GWPs. Agents must be capable of distinguishing between valid and invalid hand patterns to achieve a high level of performance.In addition, the various scoring patterns of MCR render the environment multi-goaled. Although most patterns comply with the GWPs, some special patterns do not. Notably, in many situations, these special patterns can be the closest and easiest goals to pursue. These special patterns add to the diverse choices of goals other than GWPs and thus require effective exploration by agents.

Besides, the state transitions of Mahjong can be approximately represented by a directed acyclic graph. To win a game, agents are expected to make around 10 to 40 consecutive decisions. Mistakes or poorly sampled actions in Mahjong can lead to much worse game states and are hard to recover from, such as destroying some **melds**. Such a property of Mahjong conflicts with the need for exploration and poses additional challenges to learning-based agents. Furthermore, Mahjong involves high randomness and uncertainty from drawing tiles to opponent moves. During gameplay, newly encountered game states are rarely seen during training, and it is difficult and impractical to measure the similarity between states to draw on past experience. Thus, Mahjong predominantly presents out-of-distribution (OOD) states to its agents and imposes high demands on its agents' generalization capabilities.

### B.3 REWARD SETTING FOR MCR MAHJONG ENVIRONMENT

In MCR Mahjong environment, we implement dense rewards to encourage agents to approach a winning hand more quickly, by incorporating **Shanten Distance** to calculate the reward in each step. **Shanten Distance** measures the minimum distance between the agents' current hand and any valid winning pattern. Thus, agents receive a small positive reward by decreasing **Shanten Distance** and a small penalty by increasing **Shanten Distance**.

Additionally, MCR Mahjong environment differentiates between winning by self-drawing and winning with a tile from other players. Agents will receive higher rewards if they win by self-drawing, and other players will receive the same penalty for losing. Otherwise, it will receive a positive reward, but the player who played the last tile will receive a larger penalty to discourage reckless play. Table 7 presents the reward settings for MCR Mahjong Environment.

## C EXPERIMENT SETUP AND CONFIGURATIONS

We conducted our experiments on Intel Xeon Gold 6348 CPU@2.6GHz platform with one Nvidia GeForce 3080 GPU and 1024GB RAM. For the software platform, we use Python 3.9.16, CUDA 12.4, Pytorch 2.5.1, and PyMahjongGB 1.2.0 on Ubuntu 20.04.

Tables 8, 9, and 10 present the experimental configurations for the Blackjack, Maze, and MCR Mahjong environments, respectively. These configurations were determined via manual parameter searching and comparative analysis of results upon agent convergence. For the MPPO students in the MCR Mahjong environment, as they continue training from behavior-cloning checkpoints, their policy networks are frozen for the first 1000 GPU iterations to fit the value networks alone without breaking the policy.

Table 7: Reward Settings for MCR Environment

| Agent Event | Value |
|---|---|
| Flat Step Penalty | -0.0006 |
| Decrease in Shanten Distance | 0.07 |
| Increase in Shanten Distance | -0.07 |
| Win by Self-drawing | 0.8 |
| Win with Other Player's tile | 0.6 |
| Game Lost | -0.2 |
| Game Lost with playing the final tile | -0.5 |
| Nobody Wins | 0 |

Table 8: BlackJack Experiment Configuration

| Entry | Setting | Entry | Setting |
|---|---|---|---|
| Replay Buffer | | Iteration Per | |
| -Size | 4100 | -Model Sync | 1 |
| GAE Lambda | 0.98 | Entropy Coeff | 0 |
| Batch Size | 4096 | Entropy Decay | 1 |
| Policy Coeff | 1 | Value Coeff | 0.1 |
| Gamma | 1 | Learning Rate | 1e-2 |
| PPO Epoch | 3 | PPO Clip | 0.05 |
| Normal Actor | 76 | LfD Actor | 4 |
| Gail/Sail | | Gail/Sail | |
| -Discriminator | | -Learning Rate | 1e-5 |
| -Steps/Iteration | 8 | Run Duration | 1 hour |

Table 9: Maze Experiment Configuration

| Entry | Setting | Entry | Setting |
|---|---|---|---|
| Replay Buffer | | Iteration Per | |
| -Size | 8200 | -Model Sync | 1 |
| GAE Lambda | 0.98 | Entropy Coeff | 0 |
| Batch Size | 8192 | Entropy Decay | 1 |
| Policy Coeff | 1 | Value Coeff | 0.5 |
| Gamma | 1 | Learning Rate | 5e-5 |
| PPO Epoch | 3 | PPO Clip | 0.05 |
| Normal Actor | 75 | LfD Actor | 5 |
| Gail/Sail | | Gail/Sail | |
| -Discriminator | | -Learning Rate | 1e-5 |
| -Steps/Iteration | 8 | -Run Duration | 1 hour |

Table 10: MCR Mahjong Experiment Configuration

| Entry | Setting | Entry | Setting |
|---|---|---|---|
| Replay Buffer | | Iteration Per | |
| -Size | 4100 | -Model Sync | 1 |
| GAE Lambda | 0.98 | Entropy Coeff | 1.5e-1 |
| Batch Size | 4096 | Entropy Decay | 0.99998 |
| Policy Coeff | 1 | Value Coeff | 1 |
| Gamma | 1 | Learning Rate | 1e-5 |
| PPO Epoch | 5 | PPO Clip | 0.05 |
| Normal Actor | 70 | LfD Actor | 10 |
| Gail/Sail | | Gail/Sail | |
| -Discriminator | | -Learning Rate | 1e-5 |
| -Steps/Iteration | 5 | Run Duration | 7 Days |

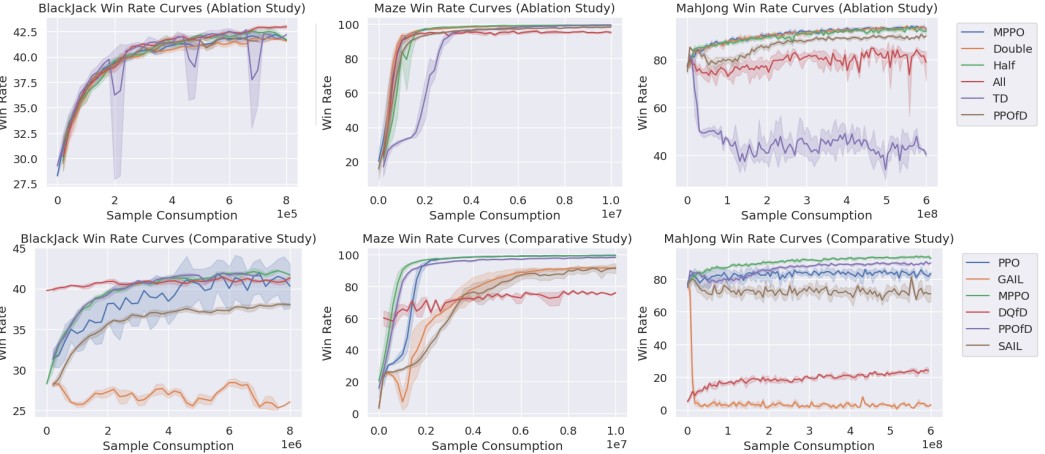

Figure 6: Win Rate Curves During Training

# D WIN RATE CURVES

Figure 6 presents the win rate curves for all experimental runs in the ablation study and comparative study. For each environment, we use Bot A to generate demonstration trajectories, which are utilized by all variants of MPPO in the ablation study and by all methods in the comparative study, except for PPO. For experiments in Mahjong environment, all methods are initialized with behavior-cloning checkpoints except DQfD, which has its own behavior-cloning pre-training phase before reinforcement learning.

It should be noted that the "win rate" metric presented in this section refers to the win rate of games generated during training, which differs from the win rates reported in the experiment section of the main text. During training, actors sample from action distributions of behavior policies, while during testing, actors always take the action with the highest logit value. In our original experiments, the sample generation rate and the sample consumption rate were recorded at fixed time intervals; however, to compare the performance of different algorithms, these metrics have been converted to accumulated sample consumption.

# E USE OF LARGE LANGUAGE MODEL

In the preparation of this manuscript, the author utilized a large language model (LLM) for the purpose of text polishing and refinement. This includes improving grammar, sentence structure, and overall clarity. The author remains solely responsible for the entire academic content, including all ideas, arguments, and conclusions presented herein.

# F COMPARATIVE ANALYSIS OF STYLE DISTANCE METRICS

We re-calculate $D_{policy}$ for BlackJack, Maze, and Mahjong environment using $W_2$ by setting $d_A(a_i, a_j) = 1, \forall i \neq j$ and $d_A(a_i, a_i) = 0$. The Pearson correlation coefficient between these two sets of aggregated distances was 0.9742, indicating a very strong positive correlation and confirming that TV divergence is a highly consistent relative measure for play style difference in our contexts.

To address the relationship between the Total Variation (TV) divergence and the 2-Wasserstein ($W_2$) distance for measuring policy difference, we conducted a correlation analysis. We recomputed the $D_{policy}$ metric for the Blackjack, Maze, and Mahjong environments using the $W_2$ distance, defining the ground metric on the discrete action space as $d_A(a_i, a_j) = 1, \forall i \neq j$. The per-state distances were aggregated into a global $D_{policy}(W_2)$ following the same expectation as in Eq. equation 2. The results are shown in Table 11, 12, and 13. The Pearson correlation coefficient between the TV-based and the $W_2$-based $D_{policy}$ across all evaluated policy pairs was 0.9742. This near-perfect positive correlation confirms that the TV divergence serves as a highly consistent and reliable relative measure for play-style difference in our contexts, justifying its use for comparative analysis in this work.

Table 11: $D_{policy}(W_2)$ results for Maze

| Agents | Optimal | Bot A | Bot B |
|---|---|---|---|
| PPO | .057±.003 | .510±.005 | .494±.004 |
| MPPO A | .084±.001 | .469±.010 | .536±.011 |
| MPPO B | .076±.004 | .536±.011 | .470±.013 |

Table 12: $D_{policy}(W_2)$ results for Mahjong

| Agents | MPPO | PPO |
|---|---|---|
| Bot A | 0.297±.019 | 0.678±.022 |
| Bot B | 0.328±.006 | 0.691±.010 |
| Bot C | 0.279±.016 | 0.772±.022 |

Table 13: $D_{policy}(W_2)$ results for BlackJack

| Agents | Optimal Policy | Bot A |
|---|---|---|
| PPO | 0.042±0.009 | 0.259±0.007 |
| MPPO | 0.135±0.004 | 0.150±0.011 |

Table 14: Mahjong MPPO A Rand. Init. Results

| Metric | Warm Start | Random Initialization |
|---|---|---|
| Win Rate | 51.05±1.43 | 51.31±1.51 |
| $D_{policy}$ | 0.297±.016 | 0.305±.010 |

# G  MAHJONG MPPO AGENT WITH RANDOM INITIALIZATION

In the main experiments, MPPO and PPO agents in the Mahjong environment were warm-started from behavior cloning (BC) checkpoints derived from their respective demonstrator bots. We clarify that this warm-start is employed solely to reduce wall-clock time and resource consumption and is not a methodological prerequisite for MPPO.

To validate this claim, we conducted an ablation study on the Mahjong environment using Bot A's demonstration data. We compared an MPPO agent initialized from a BC checkpoint against an MPPO agent initialized from random weights. All other experimental settings, including network architecture, hyperparameters, and the mixture ratio of demonstration data ($\beta = 0.05$), remained identical.

The results are summarized in Table 14. The final win rates were nearly identical, and the policy distance to the demonstrator remained low in both cases. This confirms that the demonstration data integrated via the MPPO objective is sufficient to effectively guide policy improvement and style preservation, even without a pre-trained policy. The complete win rate curve against the baseline bot and the corresponding training curve for this experiment are presented in Figure 7 and Figure 8, respectively.

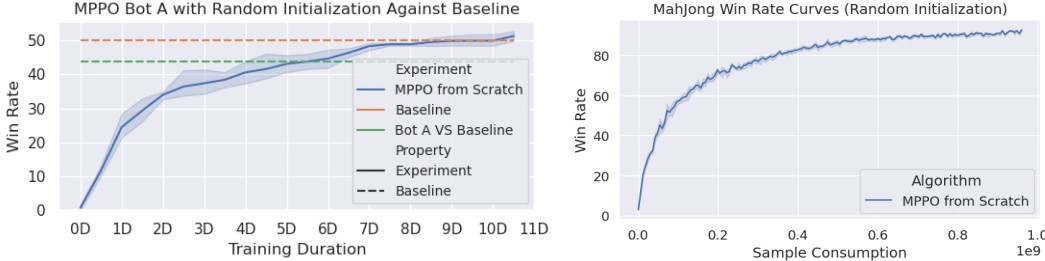

Figure 7: Win Rate Against the Baseline for MPPO Figure 8: Win Rate Curves During Training Agent A When Trained from Random Initialization. for MPPO Agent A (Random Initialization).

