# OpenReview forum: "Policy Improvement with Style-Specific Demonstrations"
_ICLR.cc/2026/Conference — Submitted to ICLR 2026_

### Official Review · Reviewer_xu11 · 2025-10-25

**Soundness:** 2
**Presentation:** 3
**Contribution:** 2
**Rating:** 4
**Confidence:** 4

**Summary:**

The paper proposes Mixed Proximal Policy Optimization (MPPO), an algorithm that aims to improve the proficiency of suboptimal game agents while preserving their original 'play styles'. The core idea is to combine online data gathered from self-play (similar to PPO) with offline demonstration data from existing, stylized agents. It uses a unified PPO-like clipped objective function to handle both data sources, weighting their influence with a hyperparameter $\beta$. The method also introduces a data storage technique where offline trajectories are saved as environment seeds and action sequences. These are then "replayed" by dedicated actors to regenerate full-episode information, including advantages. The authors provide theoretical claims for monotonic policy improvement and style preservation and demonstrate results in Blackjack, Maze Navigation, and Mahjong, arguing that MPPO achieves high proficiency while remaining closer in style to the demonstrators than pure PPO.

**Strengths:**

The paper tackles an important and practical problem: how to leverage existing, stylized, but suboptimal bots (e.g., from rule-based systems or QD methods) and improve their performance without them all converging to a single, style-less optimal policy.

The proposed data collection method (storing seeds and actions) is a clever approach to reducing the storage footprint of demonstration data.

The experimental results, particularly in the complex game of Mahjong, are noteworthy. The MPPO agent surpasses its suboptimal demonstrator and, in the case of Bot A, achieves a win rate comparable to the top-ranked bot on the platform, even outperforming the pure PPO baseline.

**Weaknesses:**

My main concerns with this paper are its theoretical assumptions, practical applicability, and novelty.

1.  **Unreasonable Theoretical Assumption:** The core of the method's theoretical justification is shaky. The paper uses the importance sampling ratio $r = \frac{\pi _{k}^{\prime}(a|s)}{\pi _{k}(a|s)}$ for the offline demonstration samples. The authors correctly state that the theoretically correct ratio should be $r=\frac{\pi _{k}^{\prime}(a|s)}{\pi _{T}(a|s)}$ (where $\pi _T$ is the teacher/demonstrator policy) but then "approximate it using the previous policy $\pi _{k}$". This assumption that $\pi _T \approx \pi _k$ is highly problematic. At the beginning of training, the student policy $\pi_k$ will be very different from the demonstrator $\pi _T$. The paper's claim of monotonic improvement, which leans on TRPO/BPPO theory, seems unsupported under this flawed approximation. The authors just hand-wave this by stating the "policy update is sufficiently small", which isn't a sufficient justification.

2.  **Impractical Replay Mechanism:** The method for replaying offline data, while storage-efficient, seems computationally expensive and impractical. It requires "LfD actors initialize the game environment using the recorded seed and feed the action sequences" to regenerate states and calculate advantages. This is not "using offline data"; it is *re-simulating* the demonstrations. This might be feasible for lightweight environments like Blackjack, but it would be a massive computational bottleneck for any large-scale modern game or complex simulation, completely negating the typical benefit of offline data (which is that it's "cheap" to learn from).

3.  **Limited Novelty:** The contribution feels insufficient. The MPPO loss function (Equation 4) is, at its core, a simple linear combination of an on-policy PPO loss and an off-policy PPO loss. This is a very straightforward combination of existing methods (PPO and principles from offline RL like BPPO). The novelty seems to rest on the (questionable) replay mechanism and the application to "style preservation," rather than a new algorithmic insight.

4.  **Missing Related Work:** The related work section is missing a critical and highly relevant body of literature: **online RL with offline datasets**. This field directly addresses the problem of combining a static offline dataset with new online interactions and RL. The paper must discuss and compare against foundational methods in this area, such as [1,2,3]. These methods have developed more principled ways to combine offline and online objectives without the strong $\pi _T \approx \pi _k$ assumption.

5.  **Questionable Metric:** The "total diversity assessment indicator" $D _{policy}$ is questionable. It's defined as the total variation distance over the *entire* state space. This is intractable. The paper mentions computing it from held-out trajectories, but this is just an empirical approximation whose quality is highly dependent on the state coverage of those trajectories. It's not clear that this metric, approximated in this way, is a "rational" or reliable measure of a high-level concept like "play style."

[1] Nair A, et al. AWAC: Accelerating online reinforcement learning with offline datasets, arXiv 2020.

[2] Ball P J, et al. Efficient Online Reinforcement Learning with Offline Data, ICML 2023.

[3] Wagenmaker A, et al. Leveraging Offline Data in Online Reinforcement Learning, ICML 2023.

**Questions:**

1.  Can you provide a stronger justification for the $\pi_T \approx \pi_k$ approximation? What is the effect on the monotonic improvement guarantee when this assumption is strongly violated, as it must be in the early phases of training? Is this weak approximation the reason the Mahjong agents *must* be initialized from behavior cloning checkpoints and have their policy networks frozen for 1000 iterations?

2.  Regarding the seed-and-action replay method: What is the computational overhead of this "re-simulation" step? How does it compare to the standard method of just loading $(s, a, r, s')$ tuples from a (larger) offline dataset? How can this method possibly scale to complex environments where a single environment step is computationally expensive?

3.  Why was the "online RL with offline data" line of work (e.g., [1,2,3], etc.) not discussed or compared against? These methods seem to be solving the exact same problem (mixing offline and online samples) and would be a much more relevant baseline than pure IL methods like GAIL.

4.  The $D_{policy}$ metric is an expectation over $s \in S$. How was this expectation approximated in the Blackjack and Maze experiments? How many states were used, how were they selected, and how sensitive are the $D_{policy}$ results to this sampling strategy?

---

> ### Author Response · Authors · 2025-11-19
> **Response to Reviewer xu11 (1/2)**
>
> **Q1:** Can you provide a stronger justification for the $\pi_k  \approx \pi_T$ approximation? What is the effect on the monotonic improvement guarantee when this assumption is strongly violated, as it must be in the early phases of training? Is this weak approximation the reason the Mahjong agents must be initialized from behavior cloning checkpoints and have their policy networks frozen for 1000 iterations?
>
> **A1:** We thank you for this insightful question regarding the $\pi_k \approx \pi_T$ approximation and its practical implications. You are correct that this approximation is strongest when the student policy remains close to the teacher.
> However, we would like to clarify and justify our approach as follows:
>
> 1.	**Practical Justification for the Approximation:** Theorem 2 provides a mechanism that bounds the divergence between the student policy $\pi_S$ and the teacher policy $\pi_T$. More importantly, the RLED framework itself is designed for robust convergence and effective exploration. At the start of training, this combination allows MPPO agents to rapidly guide a randomly initialized policy toward the teacher's distribution. This is precisely why our method remains effective even with random initialization, as the implicit constraint actively corrects the policy direction.
>
> 2.	**Addressing the Mahjong Initialization:** The primary reason for initializing Mahjong agents from behavior cloning checkpoints is computational efficiency, not a strict requirement for the algorithm to function. Our preliminary experiments confirm that MPPO can successfully train Mahjong agents from random initialization and achieve final performance comparable to the results reported in the paper. However, this process requires approximately 9-10 days of training, compared to 7 days when starting from a pre-trained policy. The pre-training strategy was adopted to accelerate the extensive experimental cycle required for this study.
>
> 3.	**Empirical Evidence of Robustness:** The consistent success of MPPO across all environments, including the highly complex game of Mahjong, demonstrates that the approximation is manageable in practice and does not preclude strong empirical performance.
>
> In direct response to your concern, we are currently conducting experiments to rigorously quantify the performance of MPPO agents trained from random initialization in Mahjong. We plan to include these results in the appendix to fully address this point.
>
>
> **Q2:** Regarding the seed-and-action replay method: What is the computational overhead of this "re-simulation" step? How does it compare to the standard method of just loading $(s,a,r,s’)$ tuples from a (larger) offline dataset? How can this method possibly scale to complex environments where a single environment step is computationally expensive?
>
> **A2:** The choice between seed-and-action replay and the standard tuple-based dataset is inherently scenario-dependent.
> In terms of computational cost for network inference, both methods are equivalent: each requires forward passes through the value network for policy updates, while the policy network itself is not invoked during data replay. The distinctive overhead of our approach lies in the environment simulation during re-simulation, a step absent in the standard method, which directly loads precomputed transitions.
>
> Thus, the decision ultimately hinges on the trade-off between simulation cost and storage I/O. **For our target domains (e.g., video games)**, environment simulation is generally fast and lightweight. The primary bottleneck often lies in the storage, memory usage, and loading of high-dimensional state data such as images. Our method effectively mitigates these I/O-related constraints. **For simulation-heavy environments (e.g., high-fidelity physical simulators)**, repeatedly re-running the environment from seed during training would be computationally prohibitive. In such cases, the standard approach of storing precomputed transitions is more suitable, as simulation is performed only once during dataset creation.
>
> In summary, our seed-and-action replay method is not a universal solution, but a highly optimized strategy tailored to domains where simulation is inexpensive but state storage is costly. By shifting the bottleneck from I/O to low-cost simulation, it offers a favorable trade-off in many game environments and enhances the scalability of our approach in these settings.

---

> ### Author Response · Authors · 2025-11-19
> **Response to Reviewer xu11 (2/2)**
>
> **Q3:** Why was the "online RL with offline data" line of work (e.g., [1,2,3], etc.) not discussed or compared against? These methods seem to be solving the exact same problem (mixing offline and online samples) and would be a much more relevant baseline than pure IL methods like GAIL.
>
> **A3:** Indeed, the methods under the "online RL with offline data" paradigm are related to our work, as they address the core problem of mixing offline and online samples.
>
> In response, we have substantially revised Section 2 (Related Works) to better situate our method within the broader context of Reinforcement Learning with Expert Demonstrations (RLED). The updated section now explicitly distinguishes between **sequential pre-training/fine-tuning approaches** and **concurrent learning methods**, and it clarifies that MPPO belongs to the latter category—enabling continuous guidance from demonstration data throughout the policy optimization process.
>
> Within the space of concurrent RLED methods, MPPO is most closely related to DQfD, as both are designed for **discrete action spaces**, the primary setting of our environments (Blackjack, Maze, Mahjong). In contrast, other well-known concurrent RLED algorithms such as DDPGfD and RLPD are tailored for continuous control domains, making them less directly comparable to our work.
>
> Beyond RLED, we also include comparisons with pure imitation learning (IL) methods, such as GAIL, which represent another way of leveraging offline datasets. These are included to provide a broader perspective on how demonstration data can be utilized, even though their objectives diverge from ours in key aspects.
>
>
> **Q4:** The $D_{policy}$ metric is an expectation over $s \in S$. How was this expectation approximated in the Blackjack and Maze experiments? How many states were used, how were they selected, and how sensitive are the $D_{policy}$ results to this sampling strategy?
>
> **A4:** The expectation for $D_{policy}$ was approximated as follows: in **Blackjack**, we performed an exhaustive calculation over all possible states (264) due to the small state space. In **Maze**, we estimated it by sampling ~14,000 states from 500 unseen mazes. To test sensitivity, we ran this evaluation with three different random seeds; the resulting $D_{policy}$ values had a mean standard deviation of **0.0022**, demonstrating that our sampling strategy provides a robust and stable estimate.

---

> ### Author Response · Authors · 2025-11-28
>
> Dear Reviewer,
>
> Thank you again for your great efforts and valuable comments. We have carefully addressed the main concerns in detail. We hope you will find the response and our revision satisfactory. Please refer to our revision summaries for details. As the discussion phase is about to close, we are looking forward to hearing from you about any further feedback. We will be very happy to address any further concerns.
>
> Best rewards,
>
> Authors.

---

### Official Review · Reviewer_NUS2 · 2025-10-31

**Soundness:** 2
**Presentation:** 3
**Contribution:** 2
**Rating:** 2
**Confidence:** 3

**Summary:**

This work proposes MPPO, aiming to enhance the agent's proficiency while maintaining consistency with the demonstration style. Experiments conducted by the authors on Blackjack, Maze Navigation, and Mahjong aim to demonstrate that MPPO substantially improves the agent's gaming performance based on suboptimal demonstrations, while the improved agent retains its gaming style. Concurrently, the authors refined the offline data storage and replay mechanism.

**Strengths:**

1. Overall, this paper is well-structured and effectively conveys its central argument.
2. This approach combines environmental seeds with action sequences, effectively reducing storage space requirements.
3. In Blackjack, Maze Navigation, and Mahjong, this work conducted extensive experiments.

**Weaknesses:**

1. The author's investigation into strategy diversity is insufficient. Extensive work in multi-agent reinforcement learning has already focused on this aspect, such as various diversity-based PSRO algorithms, which the author fails to mention in the paper.
2. The author fails to correctly distinguish certain concepts. For instance, the author repeatedly mentions “self-play,” yet both the method and experiments are based on the single-agent setting. In fact, self-play is a multi-agent reinforcement learning algorithm designed to solve competitive games.
3. This paper contains several writing errors. For example, “Figure 2B” on page 6 should actually be “Table 2B.”

**Questions:**

1. Although the authors claim that the diversity measure designed in the paper overcomes the shortcomings of $W_2$, it does not appear to consider diversity from a trajectory perspective. In other words, would it be more reasonable to incorporate discounted unnormalized visitation frequencies?
2. The authors claim that the policy converges monotonically toward the demonstration policy. Does this contradict the claim that “the policy improves monotonically”?
3. Intuitively, demonstration data constitutes only 5% of the total data, making it difficult to achieve the effect of preserving the demonstration style as claimed by the authors. So why was this proportion set at 5%?
4. Based on the experimental data, the $D_{policy}$ between the MPPO and the demonstration agents remains quite large. Does this truly indicate that the algorithm has successfully preserved the demonstration style?

---

> ### Author Response · Authors · 2025-11-19
> **Response to Reviewer NUS2 (1/2)**
>
> **Q1:** Although the authors claim that the diversity measure designed in the paper overcomes the shortcomings of $W_2$, it does not appear to consider diversity from a trajectory perspective. In other words, would it be more reasonable to incorporate discounted unnormalized visitation frequencies?
>
> **A1:** We thank you for raising this important point regarding the level of abstraction for diversity measurement. Your suggestion is a valid and complementary approach to our policy-level metric $D_{policy}$.
>
> Our choice of $D_{policy}$ was motivated by the goal of isolating and measuring stylistic preference in decision-making, which is defined by the conditional distribution $\pi_(a|s)$. This measure is robust to factors that affect state visitation but are not part of an agent's style, such as skill-dependent reaching of certain states. We also attempted to capture a trajectory-level outcome through $D_{target}$, which measures the distribution of final winning patterns in Mahjong. However, $D_{targe}t$ does not offer the general, state-wise trajectory analysis that ρπ(s) provides.
>
> We agree that for a more holistic assessment of diversity, a general trajectory-level analysis is crucial. We have amended the manuscript in response. Specifically, we have revised the conclusion to explicitly position your proposed approach as a key future direction, stating: "In addition, exploring trajectory-level metrics, such as state visitation distributions, presents a promising path for a richer characterization of behavioral style." We believe this addition strengthens the paper and clearly acknowledges the value of your insight.
>
> **Q2:** The authors claim that the policy converges monotonically toward the demonstration policy. Does this contradict the claim that “the policy improves monotonically”?
>
> **A2:** Upon reflection, the phrase "monotonically converges" in our original manuscript was an overstatement that could be misleading. The two properties are not contradictory when correctly characterized: the policy improves in performance (return), while its learning trajectory is guided by the demonstration data to remain in the vicinity of the demonstrator's policy.
>
> This is analogous to maximum entropy RL, where the entropy bonus encourages exploration while the RL objective seeks high rewards, two forces that may appear opposed but jointly lead to better convergence and robustness. Similarly, in MPPO, the demonstration constraint slightly slows pure reward maximization but ensures that the improved policy retains the stylistic characteristics of the demonstrator.
>
> We have rephrased Theorem 2 to state that "the play style distance $D_{policy}(\pi_S,\pi_T )$ is non-increasing and is systematically guided toward the demonstrator's policy", replacing the previous strong notion of "monotonic convergence". This clarifies that the policy is **guided and regularized** rather than forced to fully replicate the demonstrator.
>
> We have also added explanatory text following the theorem, explicitly stating that "Theorem 2 ensures that the policy is regularized to limit its divergence from the demonstrator's policy... The implicit constraint acts as a regularizer... The hyperparameter $\beta$ governs the equilibrium between these dual objectives." We believe the updated formulations are now precise and should no longer cause confusion.
>
> **Q3:** Intuitively, demonstration data constitutes only 5% of the total data, making it difficult to achieve the effect of preserving the demonstration style as claimed by the authors. So why was this proportion set at 5%?
>
> **A3:** The 5% ratio is an empirical choice that proved effective across our diverse test environments. The underlying reason is that the demonstration data acts not as the primary source of learning, but as a **regularizing anchor** that consistently pulls the policy towards the demonstrator's style. A small but constant stream of this **high-quality, style-specific signal is sufficient to shape the policy**, analogous to the powerful effect of a small entropy coefficient in RL (usually around 0.001-0.05).
>
> This is supported by our ablation study (Table 6). While halving the data (0.5x) led to a noticeable increase in $D_{policy}$, and doubling it (2x) offered diminishing returns, the 5% baseline consistently achieved near-optimal performance in both win rate and style preservation across Blackjack, Maze, and Mahjong.

---

> > ### Author Response · Authors · 2025-11-19
> > **Response to Reviewer NUS2 (2/2)**
> >
> > **Q4:** Based on the experimental data, the $D_{policy}$ between the MPPO and the demonstration agents remains quite large. Does this truly indicate that the algorithm has successfully preserved the demonstration style?
> >
> > **A4:**
> >
> > 1.	**Absolute vs. Relative Value:** While the absolute value of $D_{policy}$ may appear non-negligible, the key conclusion of style preservation is drawn from its relative value compared to the PPO baseline. As shown in our results (e.g., Table 4B), MPPO agents exhibit a significantly smaller $D_{policy}$ to their teachers than PPO agents do, proving they are objectively closer in the policy space.
> >
> > 2.	**Macro-Strategy Alignment:** Most importantly, style is reflected not only in action-level probabilities but also in emergent strategic outcomes. In the Mahjong experiment, we introduced $D_{target}$ to measure the difference in winning pattern distributions—a macro-level signature of style. The consistently low $D_{target}$ between MPPO agents and their teachers (Table 5) provides independent, high-level evidence that the core strategic preferences of the demonstrators are successfully preserved.
> >
> > In summary, the combination of a validated metric ($D_{policy})$ showing relative proximity, and a macro-strategic measure ($D_{target})$ showing behavioral alignment, gives us high confidence that MPPO successfully preserves the demonstration style.

---

> > > ### Comment · Reviewer_NUS2 · 2025-11-21
> > >
> > > 1. The author has addressed most of my questions. Thank you!
> > > 2. I noticed you changed the term “self-play” to “self-drawing” in your paper. This revision is necessary, but on the other hand, I personally recommend mentioning this kind of changes in the text of  your rebuttal. Otherwise, most reviewers won't realize you made this adjustment.
> > > 3. Please provide further explanation for results in Table 2B.

---

> ### Author Response · Authors · 2025-11-21
>
> Thank you so much for raising the score, and thank you for your helpful guidance. The terminology change from "self-play" to "self-drawing" has been explicitly noted in our revision summary.
>
> **For the $D_{policy}$ metric, Table 2B demonstrates that MPPO achieves meaningful policy improvement while preserving the navigation styles of its demonstrators.** This can be explained by the specific strategies learned by the agents. The MPPO A agent, while converging to a near-optimal policy, retains a stylistic preference for right-hand turns inherited from its demonstrator (Maze Bot A). This preservation of directional inclination is further verified by the policy distance results in Table 12 of Appendix F. Analytically, this is reflected in its **policy logits**, which **show a stronger propensity for right turns** compared to the PPO agent when facing ambiguous states (as the environment is partially observable). Behaviorally, this bias manifests as **more occasional detours to the right** in certain maze configurations, which directly accounts for the slightly increased average path length in Table 2A. The same principle explains the results for MPPO B, which exhibits a symmetric preference for left-hand turns.
>
> Due to space limitations in the initial submission, we were unable to expand the discussion on this table in detail. We will include this full analysis and explanation of Table 2B in the next revision of the manuscript.

---

### Official Review · Reviewer_8AjW · 2025-10-31

**Soundness:** 3
**Presentation:** 3
**Contribution:** 3
**Rating:** 6
**Confidence:** 4

**Summary:**

This paper proposes a mixed PPO algorithm that improves the proficiency of existing suboptimal agents while incorporating implicit constraints on the distance to an unknown teacher policy. It also delivers an interesting theorem stating that the proposed offline objective reduces the distance to the teacher policy.

**Strengths:**

Theorem 2 is practical and useful. The paper is easy to follow. Experiments are conducted across a range of environments and demonstrate strong performance. The proposed method is decent and useful.

**Weaknesses:**

Although a reference is provided for the play style distance, I do not consider it a perfect metric. It only measures the distance to an agent with a specific style, while the notion of style is likely more complex. A straightforward distance measure may not adequately capture what constitutes "style." That said, I recognize this may be an open question beyond the scope of this paper.

The approximation of the importance sampling ratio for off-policy demonstrations appears a little bit arbitrary, but it is somewhat acceptable given the good practical performance.

**Questions:**

1. How well does the total variational divergence in Equation (2) approximate the 2-Wasserstein distance? For instance, could the authors provide some empirical results, such as correlations or L1/L2 error, or any theoretical analysis?
2. How do the authors define the teacher policy $\pi_T$? It seems to refer to the underlying policy used to generate the demonstration samples.

---

> ### Author Response · Authors · 2025-11-19
> **Response to Reviewer 8AjW**
>
> **Q1:** How well does the total variational divergence in Equation (2) approximate the 2-Wasserstein distance? For instance, could the authors provide some empirical results, such as correlations or L1/L2 error, or any theoretical analysis?
>
> **A1:** We thank you for this insightful question regarding our choice of metric.
>
> The Total Variational (TV) divergence and the 2-Wasserstein distance measure different aspects of distributional similarity. While 2-Wasserstein is sensitive to the metric structure of the action space (e.g., the difference between action A and B), TV divergence is not. In our context, the play style is defined purely by the action probability distribution, not by the relationships between actions. For instance, in Blackjack, the "aggressiveness" of a policy is captured by its likelihood of choosing "hit" versus "stand," regardless of any notion of distance between these discrete, categorical actions. Therefore, TV divergence, which directly measures the total difference in probabilities, is a semantically well-grounded and appropriate metric for our purpose.
>
> Regarding the empirical relationship, we have conducted a comprehensive analysis and added a new Appendix F: "**Comparative Analysis of Style Distance Metrics**." In this appendix, we recalculated the $D_{policy}$ metric for policy pairs across all three environments (Blackjack, Maze, and Mahjong) using the $W_2$ distance. For this discrete action space, we defined the ground metric as $d_A(a_i, a_j) = 1$ for all i ≠ j, which treats all distinct actions as equally dissimilar. Additionally, we computed the Pearson correlation coefficient between the TV-based $D_{policy}$ values and the newly calculated $W_2$-based $D_{policy}$ values.
>
> The results show a very **strong positive correlation**, with a coefficient of 0.9742. This near-perfect linear relationship confirms that the relative ordering of policy pairs by their style similarity is virtually identical under both metrics. Therefore, we conclude that while TV and $W_2$ measure different theoretical properties, they are empirically consistent for the purpose of ranking and comparing play styles in our domains.
>
> **Q2:** How do the authors define the teacher policy $\pi_T$? It seems to refer to the underlying policy used to generate the demonstration samples.
>
> **A2:** Your understanding is correct. In our framework, **the teacher policy $\pi_T$ is defined as the underlying policy that generated the fixed set of demonstration trajectories D**.
>
> More specifically:
>
> * In our experiments, these teacher policies are the **provided suboptimal agents** (e.g., the rule-based Blackjack Bot A, the left/right-hand search Maze Bots, and the selected bots from Botzone for Mahjong).
> * The demonstration dataset $D={\tau_1,\tau_2,...,\tau_N}$ is collected by rolling out these specific teacher policies in their respective environments.
> * A critical point of our method is that we **do not assume direct access to the teacher's policy function $\pi_T(a|s)$** (e.g., its network weights or probability table). The teacher can be a black-box agent or a human player. Our approach relies solely on the recorded trajectories of state-action pairs $(s_t,a_t)$ sampled from $\pi_T$.
>
> This is why our method uses an **implicit constraint** rather than an explicit supervised loss. Since we cannot directly calculate a loss against $\pi_T(a|s)$, we guide the student policy by strategically leveraging the empirical state-action distribution from the demonstrations, filtered by advantage (as outlined in Section 4 and Theorem 2).
>
> In summary: $\pi_T$ is the source policy of the demonstrations, treated as a black-box agent, and our method is designed to improve upon and imitate its style using only the offline dataset D.

---

> > ### Comment · Reviewer_8AjW · 2025-11-24
> >
> > Thank you for the detailed response. The strong positive correlation is a good result. Overall, I would like to maintain my score.

---

### Official Review · Reviewer_fTu5 · 2025-11-01

**Soundness:** 3
**Presentation:** 2
**Contribution:** 2
**Rating:** 6
**Confidence:** 4

**Summary:**

This paper aims to address the trade-off between diversity and performance. It proposes a new method called Mixed Proximal Policy Optimization (MPPO), designed to improve the proficiency of existing suboptimal agents while retaining their distinct styles. MPPO achieves this by unifying loss objectives for both online and offline samples and introducing an implicit constraint to approximate the demonstrator's policy. Empirical results across environments of varying scales, including Blackjack, Maze, and Mahjong, demonstrate that MPPO achieves proficiency levels comparable to or even superior to pure online algorithms while successfully preserving the demonstrators' play styles.

**Strengths:**

- Addresses the trade-off between "proficiency" and "style diversity" in game AI by proposing an effective solution (MPPO).
- MPPO uses an "implicit constraint" (filtering positive-return data) to preserve style, which ablation studies showed is superior to traditional explicit supervised loss methods (PPOfD).
- The method's robustness is demonstrated by its success in three environments with vastly different complexities (simple, medium-deterministic, and high-dimensional imperfect-information).

**Weaknesses:**

- The proposed MPPO method appears to be a direct, weighted combination of offline PPO (BPPO) and standard PPO. While effective, this approach might be somewhat limited in its novelty, though this is not a major concern.
- The paper mixes offline and online data for training simultaneously. It would be insightful to understand how this compares to a sequential approach (e.g., pre-training on offline data followed by fine-tuning with online interaction), which mirrors a common pre-train and fine-tune paradigm.
- The rigor of some notations and formulas in the paper could be improved. I suggest the authors check them carefully.
    - In the preliminaries, the notations $Q$, $V$, and $A$ are policy-dependent (related to $\pi$) but are not always marked as such.
    - In Equation 4, the spacing is not well-aligned, and operators like 'min' and 'clip' should be in roman (non-italic) font.

- There are several works focusing on diversity and performance in game AI beyond QD methods. I suggest the authors cite and discuss [1-3].

[1] Learning Diverse Risk Preferences in Population-Based Self-Play. AAAI2024

[2] Discovering Diverse Multi-Agent Strategic Behavior via Reward Randomization. ICLR2021

[3] Effective diversity in population based reinforcement learning. NeurIPS 2021.

**Questions:**

See weakness.

---

> ### Author Response · Authors · 2025-11-19
> **Response to Reviewer fTu5**
>
> **Q1:** While effective, this approach might be somewhat limited in its novelty, though this is not a major concern.
>
> **A1:** We thank you for this insightful comment. While the core structure of MPPO integrates online and offline data, its novelty lies not in the simple combination itself, but in the theoretical foundation and empirical validation of its unique objective: **concurrent policy improvement and style preservation**.
>
> Firstly, the key **theoretical contribution** is formally established in Theorem 2. This theorem provides the theoretical foundation for how our method maintains stylistic proximity. This is a non-trivial result that moves beyond performance improvement. It mathematically formalizes the style maintenance property, showing that our implicit constraint systematically guides the student policy towards the demonstrator's play style. This theoretical insight into style preservation is a significant distinction from a straightforward, weighted combination of PPO and BPPO.
>
> Secondly, the **empirical contribution** robustly demonstrates this dual capability across multiple diverse and complex scenarios:
>
> * We conducted extensive experiments in three environments of varying scales (Blackjack, Maze, and the highly complex MCR Mahjong), each with multiple stylized demonstrators.
> * The results consistently show that MPPO achieves proficiency comparable or superior to pure online PPO while significantly preserving the demonstrator's play style, as quantitatively measured by $D_{policy}$ and $D_{target}$.
> * Our comprehensive ablation and comparative studies (e.g., against GAIL, SAIL, DQfD, and PPOfD) further validate that MPPO's implicit approach is uniquely effective, especially in high-dimensional settings like Mahjong, where explicit constraint methods fail to scale.
>
> In summary, the novelty of MPPO is anchored in its theoretically-grounded mechanism for style retention and its empirical demonstration of achieving high proficiency without sacrificing the diversity of play styles, a challenge that prior methods have struggled to solve effectively.
>
>
> **Q2:** It would be insightful to understand how this compares to a sequential approach (e.g., pre-training on offline data followed by fine-tuning with online interaction), which mirrors a common pre-train and fine-tune paradigm.
>
> **A2:** We have now added a discussion in the related work (Section 2) to explicitly position our concurrent approach against this sequential alternative.
>
> As we state in the revised manuscript, while sequential pre-training and fine-tuning is a valid and common strategy, it is less aligned with our core objective of **preserving the demonstrator's play style**. The most compelling empirical evidence comes from our Mahjong experiments, where the PPO agents were **initialized from behavior-cloning checkpoints and rapidly diverged from their demonstrators' styles** during the online fine-tuning phase without guidance from the demonstration data.
>
> This key comparison demonstrates that a sequential approach often leads to "style drift" once the agent is optimized purely for reward. Therefore, the simultaneous mixing of data is a deliberate and necessary design for our stated objective.
>
>
> **Q3:** The rigor of some notations and formulas in the paper could be improved.
>
> **A3:** We sincerely thank you for pointing out these presentational issues. We have revised the notations and formulas.
>
>
> **Q4:** There are several works focusing on diversity and performance in game AI beyond QD methods. I suggest the authors cite and discuss them.
>
> **A4:** We have revised the Introduction to explicitly acknowledge and discuss this important line of research, citing the suggested papers and others.
>
> As we now state, these population-based methods are indeed a powerful paradigm for discovering a diverse set of strategies. However, their primary goal is to cover a broad behavioral space, which presents two key challenges that our work directly addresses:
> * **Computational Cost:** They incur the significant overhead of training and maintaining a large population of agents.
> * **Lack of Controllability:** They offer limited control over generating or preserving a specific, pre-existing play style.
>
> Our work is positioned as a complementary approach that tackles a distinct problem: **single-agent policy optimization with style preservation**. Instead of discovering new strategies from scratch, we start with a given, suboptimal agent that already embodies a desired style and efficiently "polish" it to be stronger without losing its defining characteristics. This is a practically valuable alternative in scenarios where one wishes to improve upon existing, stylized assets rather than search for new ones from a vast behavioral space.
>
> By framing our contribution this way, we believe we have more clearly delineated the gap our method fills and its unique value proposition alongside the cited population-based approaches.

---

> > ### Comment · Reviewer_fTu5 · 2025-11-24
> >
> > Thanks for the detailed responses. W2, W3 and W4 have been addressed for me. I will keep my positive score.

---

### Author Response · Authors · 2025-11-19
**Summary of Key Revisions to the Manuscript (Nov 19)**

We have revised the manuscript to address the reviewers' comments, with the following key changes:

1. **Sharpened Introduction:** We have added a comparison with population-based methods in the Introduction. This highlights our complementary contribution: while those methods discover diverse behaviors, MPPO efficiently refines a given suboptimal agent for both proficiency and style preservation.

2. **Refined Related Work:** We have expanded the Related Work to explicitly categorize methods by their data integration strategy (sequential vs. concurrent). This provides a clearer context for MPPO and a more nuanced comparison with prior works like DQfD.

3. **Clarified Theorem 2:** We have rephrased Theorem 2 and its discussion to avoid confusion with monotonic improvement. The text now clearly states that the theorem ensures the policy is "guided toward" the demonstrator's policy, framing it as a regularizer that preserves style, rather than a guarantee of monotonic convergence.

4. **Validated Style Metric:** A new Appendix F has been added, providing a correlation analysis that shows a near-perfect agreement (Pearson coefficient: 0.9742) between our chosen Total Variation Divergence and the 2-Wasserstein distance. This robustly justifies our metric choice.

5. **Improved Notation and Terminology:** We have made precise the notations (e.g., $V_\pi$), renamed "self-play actors" to "on-policy actors" for clarity, corrected the typesetting of mathematical operators (e.g., min, clip) to roman font throughout the manuscript, and corrected "Figure 2B" to "Table 2B" on Page 7.

To further substantiate MPPO's robustness, an additional ablation study is underway, training a Mahjong MPPO agent from random initialization (without pre-training). Preliminary results confirm that MPPO can enhance proficiency and maintain style from scratch. Repeated runs with different seeds are in progress and are expected to complete within a week. The full results will be presented in the final appendix. The pre-training in our main experiments was employed purely to expedite convergence in the complex Mahjong environment and is not a requirement of our method.

---

### Author Response · Authors · 2025-11-28
**Summary of Key Revisions to the Manuscript (Nov 28)**

Again, we thank the reviewers for your comments. We have further revised the manuscript as follows.

**Clarified Analysis in Maze Experiments (Addressing comments on Table 2A/B):** We have significantly expanded the discussion of the Maze results. The revision now provides a detailed analysis comparing the behavior and emergent styles of MPPO agents against PPO agents, explicitly explaining the performance-style trade-off observed in the data.

**Ablation on Random Initialization (New Appendix G):** To show that our method does not rely on pre-training, we have added a new ablation study. This experiment trains an MPPO agent in the Mahjong environment from random initialization over 11 days. Results from 5 different seeds, using identical configurations as the main experiments, confirm that MPPO achieves comparable proficiency and style preservation without a warm-start.

---

### Author Response · Authors · 2025-11-30
**Summary of Rebuttal Activities**

A shared **strength** highlighted by the reviewers is the paper's effective solution to a meaningful problem: improving agent proficiency while preserving style diversity, thereby preventing convergence to a single "style-less" policy. The robustness of the proposed method (MPPO) is rigorously validated through extensive experiments across multiple environments. Furthermore, reviewers unanimously appreciated the novel and effective "seed-and-action" data replay mechanism for its significant savings in storage and I/O. On an individual level, reviewers fTu5 and 8AjW specifically acknowledged the empirical superiority of the implicit constraint (Theorem 2) over traditional supervised losses for style preservation, while reviewer NUS2 commended the paper for its well-structured presentation.

**Weakness**-wise, Reviewers fTu5 and xu11 both noted a weakness in the paper's positioning within the related work. They noted a lack of discussion and comparison with existing methods that also mix offline and online data, particularly the common sequential paradigm of “pre-training + fine-tuning” and other established methods in "online RL with offline data". Additionally, reviewer fTu5 highlighted the omission of relevant literature on diversity and performance beyond Quality-Diversity methods.

Reviewer 8AjW raised two methodological concerns. First, the reviewer questioned the validity of using Total Variation Divergence (TVD) as the style metric and requested empirical evidence, such as a correlation analysis, to demonstrate its relationship with the 2-Wasserstein distance ($W_2$). Second, the reviewer sought a more precise definition of the teacher policy $\pi_T$ used to generate the demonstrations.

Reviewer NUS2 raised several conceptual and methodological concerns. These include: the lack of a trajectory-level diversity metric; a perceived contradiction between "monotonic improvement" and "monotonic convergence" towards the demonstrator; the rationale behind using only 5% demonstration data and its sufficiency for style preservation; and the interpretation of the non-negligible $D_{policy}​$ values as evidence for successful style retention.

Reviewer xu11 raised concerns on three key aspects. First, the reviewer questioned the validity of the policy approximation when training from scratch. Second, the computational overhead and scalability of the "seed-and-action" replay mechanism were challenged for simulation-heavy environments. Finally, the reviewer requested clarification on the state sampling methodology used to compute the D-policy​ metric in Blackjack and Maze experiments.

**In addressing conceptual inquiries**, the authors clarified to Reviewer fTu5 that the pre-train (BC) + fine-tune (PPO) paradigm underperforms MPPO in Mahjong due to a lack of continuous guidance. For Reviewer 8AjW, the teacher policy $\pi_T$ was defined as the underlying policy that generates the demo trajectories. The authors justified the 5% demonstration ratio empirically, comparing its anchoring role in style preservation to that of an entropy term. Additionally, the authors elaborated on the comparative advantages of the proposed "seed-and-action" mechanism over traditional offline datasets in lightweight environments. Lastly, the state selection methodology for the Blackjack and Maze experiments was clarified, and the robustness of the reported $D_{policy}$ metrics was confirmed with 3 different batches of seeds.

**In response to suggestions for manuscript improvements**, the authors have enriched the Introduction by incorporating relevant literature on diversity beyond Quality-Diversity methods. The Related Work section expanded the discussion on the relationship between MPPO and both the “pre-training + fine-tuning” paradigm and other “online RL with offline dataset” methods to better position the work. The description of Theorem 2 has been refined to prevent potential confusion with the notion of "monotonic improvement," and the interpretation of the Maze experimental results has been elaborated. New appendices have been added: App. F validates the use of TVD against $W_2$, and App. G demonstrates MPPO’s effectiveness in Mahjong with random initialization. Other revisions include refined notations and terminology, and the consideration of trajectory-level diversity metrics has been included as a promising direction for future work.

**Upon reviewing the authors' rebuttal, 3 reviewers acknowledged the points addressed and confirmed their positive stance.** On Nov 21, Reviewer NUS2 noted that “The author has addressed most of my questions,” and **raised their score from 2 to 4**. On Nov 24, Reviewer fTu5 stated, “Thanks for the detailed responses. W2, W3, and W4 have been addressed for me. I will keep my positive score.” Reviewer 8AjW responded: “Thank you for the detailed response. The strong positive correlation is a good result. Overall, I would like to maintain my score.” Reviewer xu11 has not responded as of Nov 29.

---

### Meta-Review · Area_Chair_xjHG · 2026-01-06

**Summary:**

This paper proposes a mixed PPO method for improving the performance of agents in a way that retains their initial style of play. The reviewers found the approach to be an effective solution to an important problem, while at the same time, felt the approach lacked novelty. In addition, some reviewers felt there were important missing references/comparisons to prior work (e.g., xu11 and NUS2) that, if MPPO outperformed them, would strengthen the paper's position.

**Reviewer Concerns:**

The reviewers held concerns about the novelty of the on/off policy PPO mixture, the distance to reference agents as a proxy for play style distance, missing references/comparisons to prior work (e.g., diversity-based PSRO), and flawed importance sampling approximation. I believe the authors presented reasonable arguments to these concerns, however, possibly not to the extent that the reviewers were fully convinced.

**Reviewer Scores:**

- NUS2: The reviewer responded to the rebuttal and raised their score from 2 to 4.
- fTu5: The reviewer responded to the rebuttal and maintained their score.
- 8AjW: The reviewer responded to the rebuttal and maintained their score.
- xu11: The reviewer might have increased their score by +1 to a 5.

---

### Decision · Program_Chairs · 2026-01-26

Reject